# Nano-SiO_2_ and Silane Coupling Agent Co-Decorated Graphene Oxides with Enhanced Anti-Corrosion Performance of Epoxy Composite Coatings

**DOI:** 10.3390/ijms222011087

**Published:** 2021-10-14

**Authors:** Guangjie Hu, Yuxuan Xiao, Jie Ying

**Affiliations:** School of Chemical Engineering and Technology, Sun Yat-sen University, Zhuhai 519082, China; hugj5@mail.sysu.edu.cn

**Keywords:** graphene oxide, epoxy coating, surface decoration, anti-corrosion

## Abstract

Coatings are of great significance for irons and steels in regards to the harsh marine environment. Graphene oxides (GO) have been considered as an ideal filler material of epoxy coating. However, the undesired dispersion in the epoxy together with easy agglomeration and stacking remain great problems for practical application of GO composited epoxy coatings. A method that can effectively solve both self-aggregation and poor dispersion of GO is highly desired. Herein, we present a high dispersion strategy of graphene oxides in epoxy by co-decoration of nano-SiO_2_ and silane coupling agent. The co-decorated GO filled epoxy coating exhibits high anti-corrosion performance, including high electrochemical impedance, high self-corrosive potential, low self-corrosive current, and superior electrochemical impedance stability for ten days to Q235 carbon steel. This work displays new possibilities for designing novel coating materials with high performance toward practical marine anti-corrosion applications.

## 1. Introduction

Owing to the harsh marine environment, such as high salts, high oxygen, large number of marine organism, etc., irons and steels tend to go through heavy corrosion, leading to huge financial losses in regards to the development of the marine economy [1,2,3]. To address this problem, protective coatings are often applied to protect metals against corrosion. Epoxy is widely used in the marine anti-corrosion field due to its high chemical resistance, outstanding toughness, good barrier properties, high adhesion to the metallic surfaces, high durability, and other advantages [4,5,6]. However, epoxy also has its own flaws that limit its further application, including high brittleness, low tenacity and thermal shock resistance, and poor friction and wear properties [7,8,9]. Moreover, micropores, microcracks, and cavities are inevitably produced by solvent evaporation during the curing process of epoxy, which leads to the more severe penetration of oxygen, water, and salts [10,11]. To overcome these shortcomings, various inorganic fillers, such as nanocarbons [12,13], oxides [14,15,16,17], and clays [18,19], have been studied to enhance corrosion resistance. Among these filler materials, graphene oxide (GO) has drawn a great deal of attention owing to its inherent large specific surface area, good water and oxygen resistance, ionic impermeability, and chemical stability [20,21,22]. Moreover, the addition of GO could increase the electrolyte diffusing length and reduce the micropores and cavities [23]. Thus, GO is considered as an ideal filler material for epoxy coating. To synthesize high-performance GO nanosheets, different improved Humor’s methods have been developed [24,25,26]. However, it should be addressed that the high surface oxidation degree leads to the rich surface hydrophilic groups and high polarity of GO, which will make GO incompatible with many non-polar solvents and polymers [27]. On the other hand, the disadvantage of easy agglomeration and stacking will result in the undesired dispersion in the epoxy coating matrix, which greatly reduces the corrosion resistance [28,29]. Thus, there still remains big problems for the practical application of GO as a filler material.

To enhance the dispersion and anti-corrosion performance, various organic and inorganic materials are used to modify GO for dispersion enhancement. For example, nano-sized SiO_2_ [30,31], CeO_2_ [32,33], and Al_2_O_3_ [34,35] can help to enhance the space steric hindrance, avoid agglomeration and stacking, thus enhance the dispersion of GO with epoxy. However, owing to the low compatibility of these inorganic nanoparticles with epoxy, it may lead to secondary aggregation of the filler materials [36]. Using silane coupling agents is another effective method. Silane coupling agents can improve the properties of composites by combining inorganic materials with polymer materials through physical and/or chemical interactions [37,38]. Despite the fact that silane coupling agents can help to connect GO with epoxy, the agglomeration and stacking of GO still remain big problems. Thus, a method that can effectively solve both self-aggregation and poor dispersion of GO is highly desired.

Herein, a high dispersion strategy of GO nanosheets in epoxy by nano-SiO_2_ and silane coupling agent methacryloxy propyl trimethoxylsilane (KH570) co-decoration is presented. Experimental results indicate that nano-SiO_2_ and KH570 co-decorated GO (denoted as G-S-K) effectively enhance the surface hydrophobicity of GO, improve the dispersion of GO and avoid the formation of micropores in epoxy. Applied in electrochemical anti-corrosion tests, the G-S-K filled epoxy (denoted as EP-G-S-K) coating exhibits high electrochemical impedance and superior anti-corrosion durability for Q235 carbon steel. The anti-corrosion mechanism of EP-G-S-K coating has also been fully explored.

## 2. Results and Discussion

### 2.1. Synthesis and Morphology Characterization of Decorated GO

Figure 1 depicts the synthetic approach of G-S-K. GO with rich oxygen groups was first synthesized by oxidation and exfoliation of graphite (Figure 1a,b). Then, tetraethyl orthosilicate (TEOS) was used as precursor to decorate nano-SiO_2_ on GO (GO-SiO_2_, denoted as G-S), which can effectively increase the space steric hindrance and avoid the stacking of GO (Figure 1c). Finally, G-S-K was obtained by further connecting silane coupling agent KH570 with GO, which will highly improve the compatibility of G-S and epoxy resin (Figure 1d).

Scanning electron microscopy (SEM) and energy dispersive X-ray spectroscopy (EDX) are used to characterize the morphology and component of these as-synthesized samples. KH570-grafted GO (GO-KH570, denoted as G-K) was also synthesized for comparison. As shown in Figure 2a, GO exhibit the morphology of stacking nanosheets owing to their high specific surface energy, which is in accordance with the previous reports [39,40]. However, such a stacking effect is not favored in regards to the dispersion of GO in epoxy and the barrier to oxygen, water, and chloride ions. The morphology of G-K is similar to GO (Figure 2b). This is because the hydroxyl and carboxyl groups are mainly distributed along the edge, while the epoxy group is in the middle of GO [41], which is also shown in Figure 1b. When decorating with KH570, most of KH570 can only react with the hydroxyl and carboxyl groups at the edges of GO. Moreover, KH570 only has three short siloxy groups (Figure 1d), thus, it only has limited stretching effect to GO. In contrast, by decorating with nano-SiO_2_, GO can be stretched effectively owing to the four long siloxy groups formed by self-condensation of TEOS with higher space steric hindrance than the three short siloxy groups of KH570. As a result, G-S exhibited distinct segregate lamellar structure with the thickness of ~200 nm and crosswise size of ~1 μm (Figure 2c). After further decorating with KH570, G-S-K shows similar segregate nanosheet morphology with G-S. As can be seen from EDX results in Figure 2a–d and Table 1, all the samples contain C and O elements, which is originated from GO. Besides, G-K, G-S, and G-S-K contain Si element with uniform distribution, which is derived from KH570 and/or TEOS, indicating the successful graft of KH570 and/or nano-SiO_2_.

### 2.2. Structural Characterization of Decorated GO

Figure 3a shows the Fourier transform infrared spectrometry (FTIR) results of GO and decorated GO. As can be seen, GO exhibit obvious characteristic absorption peaks at 3612, 1721, and 1379 cm^−1^, corresponding to the stretching vibration peak of -OH, C=O stretching vibration peak and bending vibration peak of C-OH, respectively, which proves the existence of hydroxyl and carboxyl groups in GO [42]. For G-K, G-S, and G-S-K, the characteristic absorption peaks at 3433, 1379, and 1087 cm^−1^ are corresponding to the tensile vibration peak of -OH, the bending vibration peak of C-OH and the asymmetric tensile vibration peak of Si-O-Si, respectively [42]. Moreover, both G-K and G-S-K display the antisymmetric stretching peak of -CH_2_ and C=O tensile vibration peak at 2958 and 1721 cm^−1^ [43]. All these results further prove the successful graft of KH570 and/or nano-SiO_2_.

X-ray diffraction (XRD) was conducted to characterize the crystal structure of these decorated GO. As shown in Figure 3b, the characteristic peak of GO appears at 10.97° with a layer spacing of 0.81 nm, which is much larger than the graphite powder layer spacing (0.33 nm) owing to the contained hydroxyl, carboxyl, and epoxy groups in GO. Two peaks at 21.66° and 42.06° also appear for GO, which demonstrates the samples prepared by graphite powder are a mixture of monolayered, bilayered, or multilayered GO sheets. A wide peak appears at 23.18° for G-K, G-S, and G-S-K, illustrating the presence of amorphous SiO_2_ in these samples. Meanwhile, the characteristic peak of GO at 10.97° disappeared for G-K, G-S and G-S-K, which can be ascribed to the decoration of nano-SiO_2_ and/or KH570 on the GO surface [44]. Notably, only G-K retains the peak of GO at 42.06°. This is because the layer spacing of GO at 42.06° is very small, which makes KH570 unable to enter the GO interlayer for graft reaction. However, when TEOS reacted on the surface of GO, with the increase of SiO_2_ formation, the surface energy of GO will decline to weaken the aggregation tendency of GO. This in turn results in the increase of layer spacing, facilitating the reaction of TEOS inside GO and finally the separation of GO sheets. As a consequence, the peak of 42.06° could not be detected in both G-S and G-S-K.

The Raman spectra of the samples are displayed in Figure 3c. Two peaks around 1340 and 1590 cm^−1^ represent the D and G peaks of GO, G-K, G-S, and G-S-K [45]. The enlarged Raman spectra inset of Figure 4c shows the G peak position for GO, G-K, G-S, and G-S-K are 1592, 1599, 1605, and 1599 cm^−1^, respectively. The right shift of G peak demonstrates the decrease of layer number for GO, proving both graft of KH570 and SiO_2_ can reduce the aggregation tendency of GO [46].

Although FTIR spectra, XRD patterns, and Raman spectra have proved the successful graft of KH570 and/or nano-SiO_2_, the interaction of KH570 and/or SiO_2_ with GO surface is not clear. Thus, X-ray photoelectron spectroscopy (XPS) was conducted to characterize the chemical bonds in the decorated GO. As can be seen, Si element was introduced into G-K due to the graft of SiO_2_ and/or KH570 (Figure 4a). Figure 4b shows the C 1s XPS spectra of GO, and the peaks at 284.8, 286.9, and 289 eV correspond to C-C/C=C, C-O, and O-C=O bonds, respectively, proving the existence of hydroxyl, carboxyl, and epoxy groups in GO, which is consistent with the FTIR results. Figure 4c–e are the Si 2p spectra of the decorated GO. As can be seen, the violet peaks at 102.0, 103.4 and 102.6 eV are corresponding to the Si-O-Si bonds originated from the nano-SiO_2_ and/or KH570, while the green peaks at 102.5, 103.9, and 103.1 eV can be correlated to the formation of strong O-Si-C bonds between nano-SiO_2_ and/or KH570 with GO nanosheets for G-K, G-S, and G-S-K. The formation of chemical bonds proves the successful decoration of nano-SiO_2_ and/or KH570 to GO. Moreover, the binding energy of both Si-O-Si and Si-O-C peaks for G-S-K exhibited a 0.6 eV higher shift to G-K and a 0.8 eV lower shift to G-S, respectively, which illustrates the electron accumulation/depletion effect of G-S-K is between G-K and G-S, proving the graft of both nano-SiO_2_ and KH570 for G-S-K.

### 2.3. Modification Effect Analysis of Decorated GO

To study the effect of surface modification to GO, oil-water mixing and contact angle experiments were conducted. Figure 5a shows the changes of GO, G-K, G-S, and G-S-K after mixing with water and xylene. It can be seen that GO and G-S dispersed in the water, while G-K and G-S-K dispersed in the xylene initially, indicating the hydrophilic properties of GO and G-S and hydrophobic properties of G-K and G-S-K. After vibration, GO, G-S, and G-S-K gradually returned to their initial oil/water layer, while G-K was extracted from oil to water layer, illustrating better lipophilicity than hydrophilicity of G-K. Water contact angle tests further prove the molecular polarity of these decorated GO. GO, G-K, and G-S exhibited obvious hydrophilicity with a contact angle less than 90°, while G-S-K showed a stable hydrophobic contact angle of ~130° (Figure 5b). This illustrates by co-decorating with both nano-SiO_2_ and KH570, G-S-K obtained stable surface hydrophobic properties, which will effectively prevent the corrosion of the marine environment.

### 2.4. Morphology Characterization of Decorated GO-Filled Epoxy Coating

To investigate the performance of these decorated materials as fillers, GO, G-K, G-S, and G-S-K were mixed to epoxy, forming GO epoxy coating (denoted as EP-GO), GO-KH570 epoxy coating (denoted as EP-G-K), GO-SiO_2_ epoxy coating (denoted as EP-G-S), and EP-G-S-K coating for Q235 carbon steel, respectively. Pure epoxy was also tested for comparison. SEM characterizations were conducted to show the fracture surface of coated carbon steel. As can be seen, obvious holes appear in the fracture surface of epoxy coated steel (Figure 6a), which is owing to the lack of barrier effect of filler. Thus, bubbles in epoxy can float to the surface of the coating, leaving obvious hole channels. The existence of pores will lead to the fast penetrance of water molecules, oxygen, and chloride ions, causing corrosion to the metal. Although GO and GS were added as fillers, holes are still witnessed on the fracture surface of EP-G and EP-G-S coatings (Figure 6b,d), due to the poor dispersion of GO and GS in epoxy caused by the poor compatibility of epoxy with GO and GS as well as the stacking of GO. As a result, the bubbles in the coating can aggregate and form hole channels, which could not isolate steel from external conditions effectively. After grafting of KH-570, only a few holes and some dents left by bubbles appear on the fracture surface of EP-G-K coated steel due to the coupling effect of KH570 to epoxy and GO (Figure 6c). However, bubbles still tend to aggregate, which can be speculated by the appeared dents on the coating. Optimally, with decoration of both SiO_2_ and KH-570, only slight surface dents were seen for the EP-G-S-K coating, which could provide the best barrier effect to steels (Figure 6e).

### 2.5. Anti-Corrosion Performance of Decorated GO-Filled Epoxy Coating

Figure 7a shows the electrochemical impedance spectra (EIS) of epoxy, EP-GO, EP-G-K, EP-G-S, and EP-G-S-K-coated carbon steel, respectively. The Bode spectra are displayed in Figure 7b, and the corresponding low-frequency range impedance values are recorded in Table 2. Low-frequency range impedance is the reliable index to judge the barrier property of coating, thus the barrier property of coating to corrosive medium can be analyzed by the change of low-frequency range impedance [47]. It can be seen from the Bode spectra and the data in Table 2 that the low-frequency range impedance of epoxy coating is 2.5 × 10^6^ Ω cm^2^, while that for EP-GO, EP-G-K, EP-G-S, and EP-G-S-K coatings are 1.6 × 10^8^, 1.0 × 10^8^, 1.1 × 10^9^, and 9.7 × 10^8^ Ω cm^2^, respectively. The much higher low-frequency range impedance indicates that the addition of GO, G-K, G-S, G-S-K as fillers can effectively improve the barrier performance of the epoxy coating. This can be ascribed to the enhanced space steric hindrance brought by nano-SiO_2_ and/or coupling effect KH570, which reduces the agglomeration of the GO nanosheets and promotes the dispersion of the GO in the epoxy. The EIS and Bode spectra of carbon steel after coating for ten days are also tested. As can be seen in Figure 7c, EP-G-S-K-coated carbon steel exhibited the largest electrochemical impedance after ten days, illustrating the best prevention effect of G-S-K. The Bode spectra in Figure 7d and Table 2 show that the low-frequency range impedance values of all the coated carbon steels decrease a lot after ten days. However, the low-frequency range impedance value of EP-G-S-K coated carbon steel is 1.9 × 10^6^, which is much higher than that of epoxy (1.3 × 10^3^), EP-GO (3.4 × 10^4^), EP-G-K (7.6 × 10^3^), and EP-G-S (4.5 × 10^4^), further proving the best anti-corrosion performance of EP-G-S-K coating. Figure 7e shows the equivalent electrical circuits of all samples, while R_s_, R_c_, R_ct_, Q_c_, and Q_dl_ represented the solution resistance, pore resistance, charge transfer resistance, coating capacitance, and double layer capacitance, respectively. Model I represents the equivalent electrical circuit at initial stage, and the corrosive medium did not reach the metal surface. Once the corrosive medium reached the substrate, the protection function of epoxy coating was sharply weakened and the corrosion reaction was occurred on the surface of substrate (Model II). However, for EP-G-S-K, the addition of nano-SiO_2_ and KH570 can effectively delay the time for corrosive medium to reach the substrate, thus exhibiting best anti-corrosion performance.

The anti-corrosion performance was further studied by potentiodynamic polarization curves. Generally, the electrochemical corrosion resistance of coatings can be evaluated by calculating the self-corrosive potential (E_corr_) and self-corrosive current (i_corr_) of coatings. A high E_corr_ means a good resistance to electrochemical corrosion, and a low i_corr_ indicates low corrosion rate [48]. As shown in Figure 8a, EP-G-S-K-coated carbon steel exhibits the highest E_corr_ of −0.214 V, which is much higher than that of epoxy (−0.764 V), EP-GO (−0.731 V), EP-G-K (−0.718 V), and EP-G-S (−0.290 V)-coated steels. Moreover, it can be also witnessed that the i_corr_ of EP-G-S-K is the lowest, demonstrating the best anti-corrosion performance of EP-G-S-K coating. The anti-corrosion mechanism of the coatings to Q235 carbon steels are depicted in Figure 8b–e. Without any fillers, small bubbles in the epoxy coating tend to aggregate and grow to form perforative pore channels, which could not effectively prevent carbon steel from the external corrosive medium such as water, oxygen, chloride ions, etc. (Figure 8b). As a result, the carbon steel will easily suffer from heavy corrosion. When GO nanosheets are added as the filler, they will stack together owing to their inherent thin nanosheet structure, resulting in the bad dispersion in epoxy, which cannot effectively protect carbon steel (Figure 8c). The addition of G-S can suppress the formation of perforative pore channels to some extent, but the dispersion of G-S is still not good enough (Figure 8d). By decorating with both nano-SiO_2_ and KH570, GO nanosheets can be well stretched owing to the graft of long chain groups, thus highly dispersed in epoxy to prevent the formation of pore channels, providing strong barrier effect to carbon steels (Figure 8e).

## 3. Materials and Methods

### 3.1. Materials

Graphite powder, cerium nitrate, KH570, N,N-dimethylformamide (DMF), glacial acetic acid, sodium hydroxide, anhydrous ethanol, epoxy (E-51), and polyamide curing agent (650) were purchased from Shanghai Macklin Biochemical Co., Ltd., Shanghai, China. Cetyl trimethyl ammonium chloride (CTAC) and TEOS were purchased from Aladdin Biochemical Polytron Technologies Inc. Dimethyl benzene and concentrated sulfuric acid were purchased from Sinopharm Chemical Reagent Co., Ltd., Shanghai, China. Potassium permanganate and hydrogen peroxide were purchased from Xilong Scientific Co., Ltd., Guangzhou, China.

### 3.2. Preparation of GO

A total of 2.0 g graphite powder, 7.5 g cerium nitrate, and 70 mL concentrated sulfuric acid were added to a 250 mL beaker. After stirring and ultrasonication for 30 min, 7.5 g potassium permanganate was added to the beaker with continuous stirring for 2 h in ice bath to keep the temperature below 10 °C during the whole process. Then, the beaker was kept at 50 °C with stirring for 10 h. After that, 5 g/L potassium permanganate solution was slowly dropped to the beaker and kept stirring for 1 h. Followed by continuous stirring at 90 °C for 30 min, the reaction was terminated by dropping 30% hydrogen peroxide until no bubbles occur and dropping 10 mL 35% hydrochloric acid. The product was obtained by centrifugation at 8000 r/min for 3 min, washed several times, and finally dried at 60 °C overnight.

### 3.3. Preparation of G-S

A total of 0.1 g GO powders were first dispersed in 300 mL DI water and ultrasonicated for 30 min. The pH of GO suspension was adjusted to 11–12 by adding 1 M NaOH, followed by adding 2.0 g CTAC and stirred at 55 °C for 1 h. Then, 26 g TEOS solution (6 g TEOS dissolved in 20 g ethanol) was dropped to the suspension for 1 h and stirred for 20 min. After centrifugated at 8000 r/min for 3 min, a concentrated sulfuric acid/ethanol solution (1:10, mass ratio) was then added. Finally, the product was obtained by centrifugation, washed with DI water for 3 times, and dried at 60 °C overnight.

### 3.4. Preparation of G-K and G-S-K

In a typical synthesis of G-K, 0.2 g GO powders were dispersed in 60 mL DMF and ultrasonicated for 1 h. The pH of GO suspension was adjusted to 3.5–5.5 by adding glacial acetic acid, followed by stirred at 85 °C for 1 h. Then, KH570 solution (1.0 g KH570 dissolved in 20 mL ethanol) was dropped to the suspension for 1 h. After that, the pH of the suspension was adjusted to 9–11 by adding 1 M NaOH, followed by stirred at 85 °C for 2 h. Finally, G-K was obtained by centrifugated and washed with DMF for 3 times, washed with methylbenzene for 2 times, and dried at 60 °C overnight. The synthesis of G-S-K is similar to that of G-K, expect substituting GO with G-S and using DI water instead of DMF in the whole synthesis process.

### 3.5. Materials Characterization

SEM images and EDX analyses were collected using a Gemini500 scanning electron microscope. XRD patterns were recorded on a Empyrean X-ray diffractometer equipped with Cu Kα radiation. FTIR spectra were obtained by a Nicolet/Nexus 670 infrared spectrometer. Raman spectra were obtained by a InVia Qontor Raman spectrometer. XPS measurements were performed on a ESCALAB 250 photoelectron spectrometer. TGA measurements were conducted on a TG 209 F3 Tarsus thermogravimetric analyzer.

### 3.6. Oil–Water Mixing Experiment

First, GO and G-S were dispersed in DI water by ultrasonication, while G-K and G-S-K were dispersed in dimethyl benzene. A total of 20 mL of GO, G-K, G-S, and G-S-K suspension were added to 4 centrifugal tubes respectively, and then 20 mL dimethylbenzene was added to GO and G-S suspension, and 20 mL DI water was added to G-K and G-S-K suspension. The centrifugal tubes were shaken vigorously, and kept still until stratification.

### 3.7. Preparation of Epoxy Coating

Q235 carbon steel was first polished with 800, 1200, and 2000 mesh sand paper respectively, and washed with DI water and ethanol, followed by being dried at 60 °C overnight. Then, 4.0 g epoxy, 0.5 g dimethylbenzene, and 0.2 g fillers (GO, G-K, G-S, or G-S-K) were mixed and stirred for 6 h, followed by adding polyamide 650 as curing agent. After being stirred for 10 min and vacuum defoaming, the coating was covered on the surface of carbon steel and curing at room temperature for 5 days. Finally, the coating was polished by sand paper to keep the coating thickness of ~30 μm.

### 3.8. Electrochemical Tests

A three-electrode cell was used for the electrochemical measurements on a Autolab working station from Metrohm, Switzerland. Typically, a Ag/AgCl electrode was used as the reference electrode, a platinum mesh was used as the counter electrode, the coated Q235 carbon steel (area: 4.9 cm^−2^) was used as the working electrode, and 3.5 wt% NaCl solution was used as electrolyte. The EIS tests were performed by applying the frequency range of 0.01 to 1 × 10^5^ Hz and a small sine-wave distortion (AC signal) of 10 mV amplitude. The potentiodynamic polarization curves were measured with the scanning rate of 1 mV/s.

## 4. Conclusions

In summary, an effective high dispersion method of GO in epoxy was developed by co-decoration of nano-SiO_2_ and KH570. The co-decorated G-S-K exhibited excellent surface hydrophobic properties, which can effectively barrier the corrosion of water, oxygen, and chloride ions. Applied in electrochemical anti-corrosion tests, EP-G-S-K-coated carbon steel exhibited a high electrochemical impedance, a much higher self-corrosive potential, a lower self-corrosive current, together with a several magnitude higher electrochemical impedance compared to pure epoxy, GO, G-S, and G-S-K-coated Q235 carbon steel after ten days. We believe our work displays new possibilities for designing novel coating materials with high performance toward practical marine anti-corrosion applications.

## Figures and Tables

**Figure 1 ijms-22-11087-f001:**
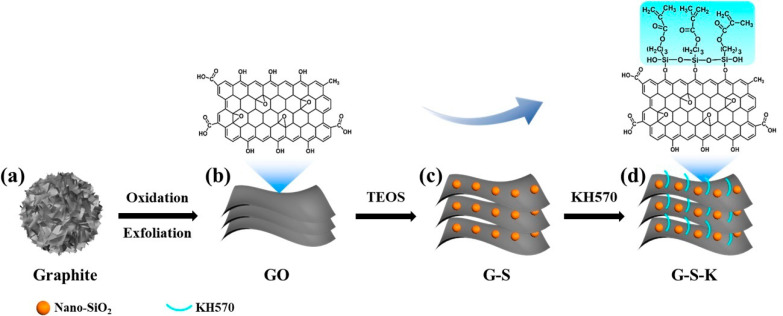
Scheme illustration of the synthesis of co-decorated GO. (**a**) Graphite. (**b**) GO and its molecular structure. (**c**) G-S. (**d**) G-S-K and its molecular structure.

**Figure 2 ijms-22-11087-f002:**
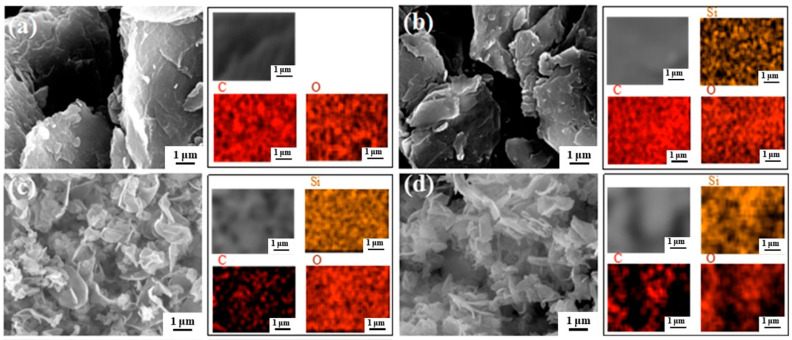
SEM images and corresponding EDX mapping results of C, O, Si for (**a**) GO, (**b**) G-K, (**c**) G-S, and (**d**) G-S-K.

**Figure 3 ijms-22-11087-f003:**
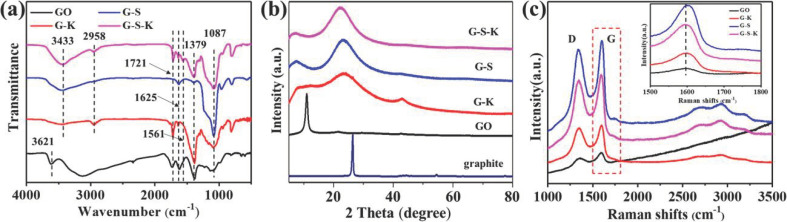
(**a**) FTIR spectra, (**b**) XRD patterns, and (**c**) Raman spectra of GO, G-K, G-S, and G-S-K.

**Figure 4 ijms-22-11087-f004:**
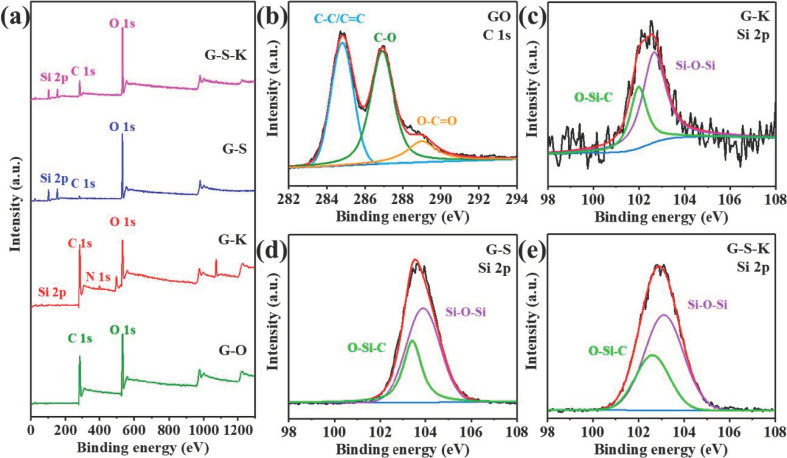
(**a**) XPS survey spectra and high-resolution XPS spectra of (**b**) C 1s signal for GO and Si 2p signal for (**c**) G-K, (**d**) G-S, and (**e**) G-S-K.

**Figure 5 ijms-22-11087-f005:**
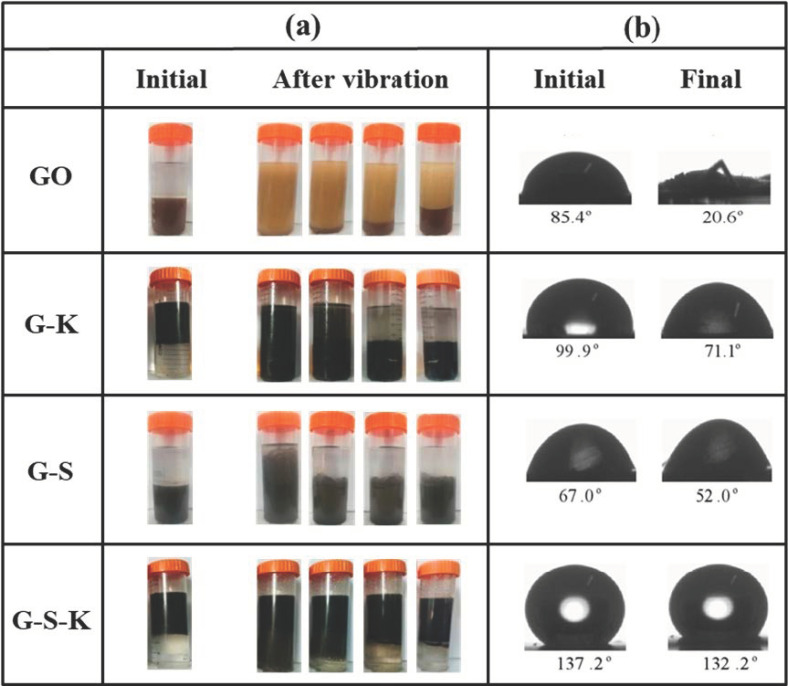
(**a**) Oil-water mixing experiment diagram and (**b**) contact angle of GO, G-K, G-S, and G-S-K.

**Figure 6 ijms-22-11087-f006:**
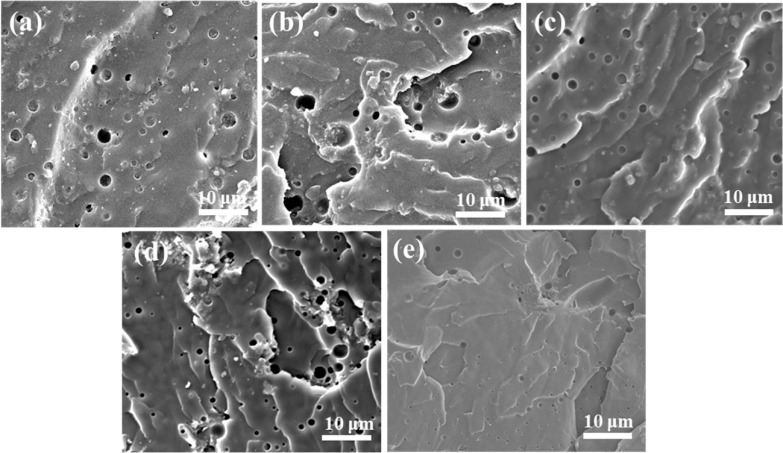
Fracture surface morphology of (**a**) pure epoxy, (**b**) EP-GO, (**c**) EP-G-K, (**d**) EP-G-S, (**e**) EP-G-S-K-coated Q235 carbon steel.

**Figure 7 ijms-22-11087-f007:**
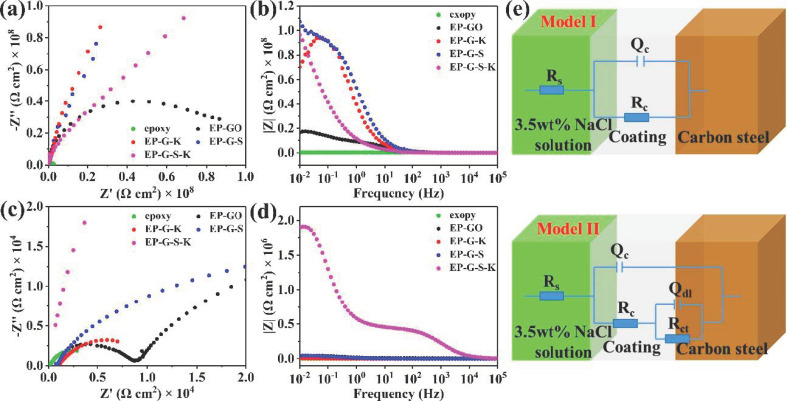
(**a**) EIS spectra and (**b**) Bode spectra of different coatings. (**c**) EIS spectra and (**d**) Bode spectra of different coatings after 10 days. (**e**) Scheme illustration of equivalent electrical circuit models for all the samples.

**Figure 8 ijms-22-11087-f008:**
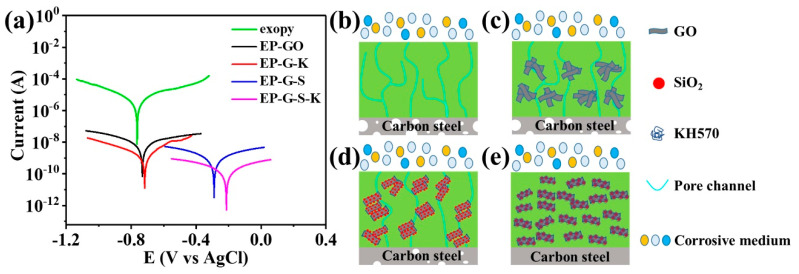
(**a**) Potentiodynamic polarization curves of epoxy, EP-GO, EP-G-S, and EP-G-S-K coatings. Schematic illustration of the barrier effect of (**b**) epoxy, (**c**) EP-GO, (**d**) EP-G-S, and (**e**) EP-G-S-K coatings to Q235 carbon steel.

**Table 1 ijms-22-11087-t001:** Element content of GO, G-K, G-S, and G-S-K from EDX results.

Element	GO		G-K		G-S		G-S-K	
	Weight (%)	Atom (%)	Weight (%)	Atom (%)	Weight (%)	Atom (%)	Weight (%)	Atom (%)
C	68.20	74.08	44.39	55.26	21.75	30.96	36.91	47.77
O	31.80	25.92	37.63	35.16	46.62	49.80	41.38	40.21
Si	-	-	17.98	9.58	31.63	19.24	21.71	12.02
Total	100	100	100	100	100	100	100	100

**Table 2 ijms-22-11087-t002:** The low-frequency range impedance |Z|_0.01Hz_ of epoxy, EP-GO, EP-G-K, EP-G-S, and EP-G-S-K before and after 10 days of immersion.

Time/day	0 (Ω cm^2^)	10 (Ω cm^2^)
epoxy	2.5 × 10^6^	1.3 × 10^3^
EP-GO	1.6 × 10^8^	3.4 × 10^4^
EP-G-K	7.0 × 10^8^	7.6 × 10^3^
EP-G-S	1.1 × 10^9^	4.5 × 10^4^
EP-G-S-K	9.7 × 10^8^	1.9 × 10^6^

## Data Availability

Not applicable.

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
