# Peer review of "Nano-SiO2 and Silane Coupling Agent Co-Decorated Graphene Oxides with Enhanced Anti-Corrosion Performance of Epoxy Composite Coatings"

_ijms, 2021, doi:10.3390/ijms222011087_

Round 1
Reviewer 1 Report
Hu et al in this manuscript reported adding of Nano-SiO2 and Silane coupling agent co-functionalized GO into epoxy can achieve better anti-corrosion performance than epoxy. The result was promising for practical application. In the control experiments, the authors further clarify the reason leading to its high performance. Therefore, I recommend for publication. However, there are some minor suggestions here for your reference.
- In table 2, why the EP-G-S shows higher resistance than EP-G-S-K at initial state since EP-G-S-K has a better anti-corrosion performance?
- The oxidation of graphene oxide is important for GO hydrophilic property as different synthetic methods show different oxidation degree. Would you be able to quantify that and explain its relationship to anti-corrosion performance ? The author can cite some papers about different oxidation degree from different synthetic methods. (Materials today Physics, 2019, 9, 100097; ACS Nano, 2010, 4,8, 4806-4814) and write a sentence or paragraph to clear that when other researchers try to use GO to repeat your work.
- Do you think 10 days are enough to check the duration of EP-G-S-K for anti-corrosion performance?
Author Response
Please find our specific point-to-point responses to the comments in the attached file.

Reviewer 2 Report
In this paper, the authors present a strategy to co-decorated GO with nano-SiO2 and silane coupling agent. The co-decorated resulting material exhibits excellent surface hydrophobic properties, which can effectively barrier the corrosion of water, oxygen and chloride ions. However, even if the work seems quite consistent, there are too many English grammar errors and misprints. To name a few:
“GO still remain problem” line 54; “eletrochemical” line 61; “seperation” line 127; “postion” line 131; “succussful” line 137; “Si element were introduced” line 140; “due to the the graft” line 141; “a few hole” line 181; “, bubbles still tends to aggreate” line 183; “efffect” line 186; “efffectively” line 201; “carbbon” line 208; “highly dispserse” line 236.
Thus, the paper needs urgently a careful revision from the authors.
Other issues:
- XPS analysis is not well deconvoluted and it is too poorly analyzed to confirm the successful chemical graft Furthermore, it is stated “The two peaks around 103 and 102 eV are corresponding to the Si-O-Si and O-Si-C bonds” but the Si-O-Si bond is not shown anywhere in Figure 4.
- What about the size of these SiO2 NPs?
- An equivalent circuit should be included in Figure 7 (the EIS measurements).
Author Response

(The authors gave the same response as above.)

Round 2
Reviewer 2 Report
The paper has been improved. However, in the XPS spectra, it can be clearly seen that for G-K sample Si-O-Si and Si-O-C position peaks are shifted with respect to G-S and G-S-K. This must be explained in the main text.
Author Response
Thank you very much for this valuable comment. We have made revision to the explanation of XPS part in the revised manuscript by yellow background as follows:
Figure 4c-e are the Si 2p spectra of the decorated GO. As can be seen, the violet peaks at 102.0, 103.4 and 102.6 eV are corresponding to the Si-O-Si bonds originated from the nano-SiO2 and/or KH570, while the green peaks at 102.5, 103.9 and 103.1 eV can be correlated to the formation of strong O-Si-C bonds between nano-SiO2 and/or KH570 with GO nanosheets for G-K, G-S and G-S-K. The formation of chemical bonds proves the successful decoration of nano-SiO2 and/or KH570 to GO. Moreover, the binding energy of both Si-O-Si and Si-O-C peaks for G-S-K exhibited a 0.6 eV higher shift to G-K and a 0.8 eV lower shift to G-S, respectively, which illustrates the electron accumulation/depletion effect of G-S-K is between G-K and G-S, proving the graft of both nano-SiO2 and KH570 for G-S-K.